# Clinical Effects of Photofunctionalization on Implant Stability and Marginal Bone Loss: Systematic Review and Meta-Analysis

**DOI:** 10.3390/jcm11237042

**Published:** 2022-11-28

**Authors:** Xinrui Lang, Bo Qiao, Ziyu Ge, Jiahui Yan, Yanzhen Zhang

**Affiliations:** Department of General Dentistry, The Second Affiliated Hospital of Zhejiang University School of Medicine, Hangzhou 310009, China

**Keywords:** dental implant, photofunctionalization, implant stability, marginal bone loss, osseointegration

## Abstract

Background: Several clinical trials have recently been conducted to elucidate the effectiveness of photofunctionalization. The aim of this review was to systematically analyze the clinical effects of photofunctionalization on implant stability and marginal bone loss (MBL). Methods: An electronic search in four databases and a manual search were conducted in September 2022. Randomized controlled trials (RCTs), clinical controlled trials (CCTs), and cohort and case-control studies evaluating the effects of photofunctionalization on implant stability or marginal bone loss (MBL) in humans were included. The methodological quality assessment using RoB 2.0 and the ROBINS-I tool was performed based on different study designs. Results: Seven studies were included for a qualitative analysis, and five of them were chosen for a meta-analysis. The meta-analysis revealed that photofunctionalization significantly improved the stability of the implant 2 months after implantation (*p* = 0.04; MD = 3.48; 95% CI = −0.23 to 6.73) and increased the osseointegration speed index (OSI) (*p* = 0.007; MD = 2.13; 95% CI = 0.57 to 3.68). However, no significant improvements of implant stability were observed 2 weeks (*p* = 0.62), 4 weeks (*p* = 0.31), nor 4 months (*p* = 0.24) after implantation. The evaluation presented no significant reductions in MBL. Conclusions: Based on the positive effect of photofunctionalization on the rate of establishing implant stability, photofunctionalization may provide an effective and practical strategy to achieve faster osseointegration and reduce the overall healing time. Photofunctionalization appears to improve the implant stability. However, the clinical effect of photofunctionalization on MBL remains unclear due to the shortage of available studies.

## 1. Introduction

Dental implants have become the preferred treatment for patients with dentition defects because of their long-term success rate, comfort, and beauty [1,2]. One of the key criteria for implant success is the formation of osseointegration, which is generally reflected by implant stability [3,4]. Moreover, implant stability is an essential prerequisite for implant survival and functional loading, and it depends on numerous factors, such as bone quality, implant design, surgical techniques, substrate type, and experience of the operator [5,6]. However, bone resorption has been considered a common phenomenon after implant placement and loading [7]. Marginal bone loss (MBL), as an important indicator for the radiological evaluation of implants, has been widely used to determine implant success and detect potential implant failure [8,9]. The criterion currently widely accepted is that the bone loss of a successful implant should not exceed 1.5 mm during the first year of function and not exceed 0.2 mm per year in subsequent years [10].

In order to enhance the osseointegration of implants, several methods of implant surface modification have been studied, and one of them is photofunctionalization, or ultraviolet (UV) treatment. It refers to a method that uses UV irradiation to modify the surface of titanium and titanium dioxide, including the alteration of physical and chemical properties and the enhancement of bioactivity, such as the improvement of efficiency and capacity for protein adsorption, the decrease in surface hydrocarbon content, and the increase in hydrophilicity [11,12,13,14]. To determine whether photofunctionalization improves dental implant osseointegration, previous studies have mostly focused on animal models. In a rat model, Aita et al. suggested that photofunctionalization enabled a more rapid and complete establishment of bone–titanium integration [15]. Bone–implant contact (BIC) was maximized up to nearly 100% on week 4 of healing. Similar findings that UV photofunctionalization increased BIC were reported in dog and rabbit models [16,17]. According to Aita et al. and Ueno et al., significantly higher BIC push-in values of photofunctionalized implants compared with the control groups were shown in rat models in the initial stage, which indicated that the osseointegration of implants was faster enhanced [15,18]. Two recent systematic reviews that concentrated on the positive effect of photofunctionalization on implant osseointegration were published, and both of them concluded that the UV treatment of titanium dental implant surfaces may be an effective strategy to improve the osseointegration process [19,20].

However, whether photofunctionalization has a positive impact on the osseointegration of dental implants in clinic is not ascertained. A few of clinical trials have recently investigated the outcomes of photofunctionalized implants, especially focused on the evaluation of implant stability and MBL [21,22,23,24,25,26,27,28,29,30,31,32]. Shah et al. showed that photofunctionalization significantly improved implant stability as compared with the control group [31]. Sandhu et al. reported that implants with photofunctionalized surfaces significantly reduced MBL [30]. However, controversial results have also been reported. For instance, no statistically significant differences in implant stability or MBL between photofunctionalized group and control group were observed by Heo et al. and Zaheer et al. [28,32]. Thus, it is essential to systematically analyze the quantitative assessment of the clinical effects of photofunctionalization in order to provide suggestions for future trials and clinical practice. To the best of our knowledge, there are no comprehensive analyses of the clinical effectiveness of photofunctionalization in current available studies. Therefore, this systematic review and meta-analysis aimed to evaluate the clinical effects of photofunctionalization on implant stability and marginal bone loss.

## 2. Materials and Methods

Our systematic review and meta-analyses were conducted in accordance with the Preferred Reporting Items for Systematic Reviews and Meta-analyses (PRISMA) guidelines [33]. The protocol of the study was registered in PROSPERO (CRD42022352186).

### 2.1. Focused Question

The focused research question was: In patients undergoing dental implant therapy, can photofunctionalization improve implant stability and reduce marginal bone loss?

### 2.2. PICOS Criteria

Patients (P): Patients, aged at least 18 years, with dentition defects, who received dental implant installations. There were no restrictions on gender.

Intervention (I): Photofunctionalization of implants.

Comparison (C): Non-photofunctionalized implants.

Outcomes (O): Primary outcomes were dental implant stability and the rate of implant stability development evaluated by calculating the osseointegration speed index (OSI), which was defined as [(ISQ at loading) − (ISQ at implant placement)]/(healing time in months) [23]. The second outcome was MBL at the mesial and the distal sides of the implant.

Study design (S): Randomized controlled trials (RCTs), clinical controlled trials (CCTs), and cohort and case-control studies.

### 2.3. Eligibility Criteria

RCTs, CCTs, and cohort and case-control studies that reported implant stability and MBL in patients (≥18 years old) receiving photofunctionalized (test group) or non-photofunctionalized (control group) dental implants were eligible for inclusion.

Animal experiments or in vitro studies, case reports, case series, reviews, comments, systemic reviews, and cross-sectional studies were excluded.

### 2.4. Search Strategy and Study Selection

Electronic searches were conducted in the databases of PubMed, Cochrane Library, EMBASE, and Web of Science without language or publication date restriction on 4 September 2022. A detailed search strategy developed for each database is available in Appendix A.

After automatically removing duplicates, two reviewers (X.L. and B.Q.) independently carried out the initial screening by reading the title and abstract. Then, the full texts of the studies qualified at the title and abstract level were read for inclusion according to the eligibility criteria. Any disagreement was discussed with a third reviewer (Y.Z.) until a consensus was reached.

Furthermore, a manual search of the relevant literature in the references of the included articles was performed. The hand-searching of the following dental implant journals was also completed up to September 2022: *Clinical Oral Implants Research*, *International Journal of Oral & Maxillofacial Implants*, *Clinical Implant Dentistry and Related Research*, *Journal of Oral and Maxillofacial Surgery*, *European Journal of Oral Implantology*, *International Journal of Periodontics & Restorative Dentistry and Implant Dentistry*.

### 2.5. Data Extraction

The following information was extracted by the same reviewers (X.L. and B.Q.) independently: first author, year of publication, country, study design, number of participants, mean age (range), the position of the implant, healing time, parameters about photofunctionalization, and outcome measurements. The corresponding author was contacted to obtain any incomplete or missing data. Any disagreement was resolved with discussion, and the third reviewer (Y.Z.) was consulted for clarification.

### 2.6. Risk-of-Bias Assessment

Two reviewers (X.L. and B.Q.) independently performed the methodological quality assessment of the selected studies according to Cochrane risk-of-bias tool 2 (RoB 2.0) for RCTs [34]. Five domains were included in the assessment of the risk-of-bias criteria, including: (1) randomization process, (2) deviations from intended interventions, (3) missing outcome data, (4) measurement of the outcome, and (5) selection of the reported result. Each domain was assigned one of three levels: low risk of bias, some concerns, or high risk of bias. The ROBINS-I tool was used to access CCTs, and cohort and case-control studies [35], which encompassed seven domains: (1) bias due to confounding, (2) bias in selection of participants into the study, (3) bias in classification of interventions, (4) bias due to deviations from intended interventions, (5) bias due to missing data, (6) bias in measurement of outcomes, and (7) bias in selection of the reported results. The response options for all domains and an overall risk-of-bias judgement were low risk, moderate risk, serious risk, critical risk of bias, or no information. Any conflict was resolved with discussion.

### 2.7. Data Analysis

Implant stability and OSI were regarded as the primary outcomes, while the secondary outcome was MBL, extracted as mean values and standard deviations (SDs). Quantitative data of implant stability, OSI, and MBL were statistically combined to calculate the mean differences (MDs) and 95% confidence intervals (95% CIs) for the meta-analysis using Review Manager 5.4 software (Cochrane Collaboration, London, UK). The level of significance was set at 0.05 (two-tailed z-tests). The Q-test was used, and the I^2^ index was calculated to assess statistical heterogeneity among the studies. If heterogeneity was found to be low (I^2^ < 50%), then a fixed effects model was employed to analyze the data, otherwise a random effects model was considered. Forest plots were generated and presented for the primary and secondary outcomes. A qualitative analysis was performed including data that were not amenable to the meta-analysis.

## 3. Results

### 3.1. Study Selection

A total of 1788 relevant articles were obtained through searching the electronic database. After 292 duplicate articles were removed, 1484 articles that did not meet the inclusion criteria were excluded based on their titles and abstracts, and 12 articles were included [21,22,24,25,26,27,28,29,30,31,32,36]. In addition, 185 articles were obtained by performing a manual search of dental implant journals and the related references of the included articles, of which three articles were subjected to full-paper review eligibility [23,37,38]. After the full text of the total 15 articles were evaluated [21,22,23,24,25,26,27,28,29,30,31,32,36,37,38], 8 studies were left for some reasons [21,22,23,25,26,36,37,38] (Appendix A). Finally, seven studies were included in the qualitative analysis [24,27,28,29,30,31,32], and five of them were chosen for the meta-analysis [24,28,30,31,32] (Figure 1).

### 3.2. Study Characteristics

Among the seven included studies, one study was published in 2016 [24], two were published in 2020 [27,28], and the other four were published in 2021 [29,30,31,32]. Five studies were RCTs [27,28,29,30,31], including two split-mouth trials [27,30], one CCT [32], and one case-control study [24]. These studies were conducted in Japan, Lithuania, Pakistan, Korea, and India, involving 430 patients and 734 implants. The implants in one study were placed in the maxillary posterior region [29] and in another one in the maxillary anterior region [31]. The other five studies included implants placed in both maxilla and mandible [24,27,28,30,32]. One study included immediate implants [31]; five studies included implants inserted after a healing period of extraction [27,28,29,30,32]; and one study included both [24]. All seven studies followed a delayed loading protocol [24,27,28,29,30,31,32], and four studies reported healing time free of prosthesis loading, ranging from 3 to 8 months [24,29,30,31] (Table 1).

With regard to the features about the photofunctionalization parameters, the devices used for UV treatment varied among the studies. Two RCTs specifically demonstrated that the wavelengths of UV radiation were 382 nm (UVA), 260 nm (UVC), and 257.3 nm [28,31]. The implants in two studies were treated with UV irradiation using a photo device, TheraBeam Affinity Device (Ushio Inc., Tokyo, Japan) [24,29], which delivered UV light as a mixture of spectra via a single source of UV lamp at λ = 360 nm and λ = 250 nm [39]. The implants in another RCT were subjected to UV irradiation using a photofunctionalization device, TheraBeam^®^ SuperOsseo Device (Ushio Inc., Hyogo, Japan) [27]. The UV light of TheraBeam^®^ SuperOsseo Device was generated as a mixture of spectra; the intensities were about 0.05 mW/cm^2^ (λ = 360 nm) and 2 mW/cm^2^ (λ = 250 nm) [40]. Moreover, Sandhu et al. performed UV light treatment using a UV machine (Lelesil Innovative Systems, Thane, India), but there was no information reported about the parameters set by the manufacturer [30]. In addition, the processing time of UV irradiation was reported to range from 10 minutes to 20 minutes in six studies [24,27,28,29,30,31]. The CCT did not report any information about the photofunctionalization device, the wavelength, or the processing time [32] (Table 2).

Hirota et al. divided implants into regular and complex placement according to host bone condition and reported that photofunctionalization accelerated the rate and enhanced the final level of implant stability development, particularly in complex cases [24]. Puisys et al. measured removal torque to assess BIC and implant stability, suggesting that the photoactivation of implants improved healing and implant stability, especially in the early healing phase [27]. Implants were respectively photofunctionalized with ultraviolet A (UVA) and ultraviolet C (UVC) irradiation in the study by Zaheer et al., and it was found that both UVA and UVC treatments reduced MBL, with no significant differences between them [28]. Choi et al. observed that UV surface treatment may increase the initial stability of dental implants in the region of the maxilla with poor bone quality [29]. Sandhu et al. showed that implants with photofunctionalized surfaces achieved faster osseointegration with good crestal bone stability and reduced crestal bone loss [30]. Shah et al. found that pretreatment with photofunctionalization exhibited a statistically significant difference in implant stability, but there were no significant differences in MBL [31]. Heo et al. also found that the photoactivated implants showed higher ISQ values than those without photofunctionalization; however, there were no significant differences [32] (Table 3).

### 3.3. Risk of Bias

The risk of bias of the five RCTs [27,28,29,30,31] evaluated with the RoB 2.0 tool revealed that one study (20%) presented some concerns [28], while three studies (60%) presented a high risk of bias [27,29,30]. The main reason for bias was the selection of the reported result. There were some concerns about baseline differences between intervention groups in two studies [28,29]. All RCTs presented a low risk of deviations from intended interventions, missing outcome data, and measurement of the outcome [27,28,29,30,31] (Figure 2).

The case-control study [24] and the CCT [32] were assessed with the ROBINS-I tool, and both were judged to present a serious risk of bias. Two studies presented moderate risk and serious risk of bias due to confounding, respectively [24,32]. There was a serious risk of bias due to missing data in the case-control study [24] (Figure 3).

### 3.4. Meta-Analyses

#### 3.4.1. Implant Stability

Six studies reported the effect of photofunctionalization on implant stability [24,27,29,30,31,32]. One study evaluated implant stability using the removal torque value [27], and the other studies used the implant stability quotient (ISQ) measured using a resonance frequency analysis (RFA) [24,29,30,31,32]. On data from studies where the ISQ measured immediately and 2 and 4 weeks, and 2 and 4 months after the placement of the implants, we conducted a meta-analysis (Figure 4).

Two studies evaluating three outcomes of the ISQ measured immediately after the placement of the implants showed that there were no significant differences between the photofunctionalization and control groups in implant stability (I^2^ = 86%; *p* = 0.38; MD = −5.73; 95% CI = −18.47 to 7.01) [24,32].

The evaluation 2 and 4 weeks, and 2 and 4 months after implant placement was performed in the studies by Heo et al. and Shah et al., including 52 photofunctionalized implants and 53 non-photofunctionalized implants [31,32]. The evaluation of the two studies showed that there were no statistically significant differences between the photofunctionalization and control groups in implant stability at 2 weeks (I^2^ = 0%; *p* = 0.62; MD = 0.24; 95% CI = −0.68 to 1.16), 4 weeks (I^2^ = 69%; *p* = 0.31; MD = 0.97; 95% CI = −0.88 to 2.82), and 4 months (I^2^ = 99%; *p* = 0.24; MD = 6.22; 95% CI = −4.19 to 16.63). However, there was a significant improvement of implant stability in the photofunctionalization group vs. control group 2 months after implant placement (*p* = 0.04; MD = 3.48; 95% CI = −0.23 to 6.73), and there was significant heterogeneity between the studies (*p* = 0.004, I^2^ = 88%).

#### 3.4.2. OSI

Two studies reported the OSI [24,30]. The evaluation of outcomes revealed that photofunctionalization resulted in a significant increase in the OSI (I^2^ = 81%; *p* = 0.007; MD = 2.13; 95% CI = 0.57 to 3.68) (Figure 5).

#### 3.4.3. MBL

Four studies reported the effect of photofunctionalization on MBL [28,29,30,31]. The mesial and distal MBLs with a follow-up time of 2 months and 6 months after the placement of the implants in two studies were selected for a meta-analysis [28,31] (Figure 6).

For the period of 2 months after implant placement, the forest plot showed high heterogeneity of the studies, and there were no statistically significant differences between the photofunctionalization and control groups in MBL both on the mesial side (I^2^ = 89%; *p* = 0.76; MD = −0.05; 95% CI = −0.40 to 0.29) and on the distal side (I^2^ = 89%; *p* = 0.29; MD = −0.22; 95% CI = −0.64 to 0.19) of the implants.

The evaluation of the outcomes at 6 months also presented no significant differences between the photofunctionalization and control groups in MBL both on the mesial side (I^2^ = 93%; *p* = 0.18; MD = −0.27; 95% CI = −0.67 to 0.13) and on the distal side (I^2^ = 97%; *p* = 0.14; MD = −0.48; 95% CI = −1.13 to 0.17) of the implants.

## 4. Discussion

The positive effect of ultraviolet-light functionalization on the osseointegration of titanium implants has received extensive attention. To the best of the authors’ knowledge, the current study is the first to elucidate the clinical effects of photofunctionalization on dental implants through the evidence-based method. In our systematic review and meta-analysis, the researchers analyzed the clinical impacts of photofunctionalization on implant stability and MBL. A total of seven studies, including five RCTs [27,28,29,30,31], one CCT [32], and one case-control study, were included [24]; finally, five studies were selected for meta-analyses [24,28,30,31,32]. The results showed that photofunctionalization significantly improved the stability of implants 2 months after implantation and increased the rate of establishing implant stability, although the improvement of implant stability 2 weeks, 4 weeks, and 4 months after the implant placement and the reduction in MBL were limited.

The ISQ values based on RFAs have been extensively used as indicators of mechanical implant stability with reasonable reliability and validity and are believed to have predictive power for clinical outcomes [41,42,43]. Five studies in this systematic review used an RFA to measure implant stability, and all of them showed the positive effect of photofunctionalization on implant stability [24,29,30,31,32]. Similarly, Suzuki et al. observed considerably higher ISQ values for immediately loaded photofunctionalized implants [23]. Significantly improved stability and successful implant osseointegration were also observed in photofunctionalized dental implants placed in complex cases requiring staged or simultaneous site-development surgery and implants with low and extremely low primary stability [22,25]. Although the abovementioned studies were excluded from our systematic review and meta-analysis after evaluating the full text because they were case series or did not present control groups, they showed promising clinical outcomes of photofunctionalized dental implants [22,23,25].

Besides RFAs, there are some different methods to assess implant stability, such as radiographical analyses, periotests, histologic/histomorphologic analyses, tensional tests, push-out/pull-out tests, removal torque analyses, etc. [44]. Removal torque, which refers to the force necessary to detach an implant from the bone, indirectly provides information on the degree of BIC [45]. Puisys et al. investigated that photofunctionalization leads to higher resistance to removal torque forces compared with non-treated implants, indicating improved healing and implant stability [27].

The results of the clinical trials are consistent with the finding in an animal study that photofunctionalization increased by three times the strength of the anchorage of the implants in the early healing stage [15]. Two existing systematic reviews summarized from available data based on animal models that photofunctionalization improved osseointegration in the initial healing period of implants [19,20]. The physico-chemical properties and biological capabilities of dental implants enhanced with photofunctionalization contributed to the outcomes [46].

The hydrophilicity of the implant surface is a key factor in the process of osseointegration [47]. After UV irradiation, the hydrophobic titanium surface becomes superhydrophilic [21,48]. The bioactivity of the titanium surface decreases with exposure time [13]. An aged titanium surface is negatively charged, but a photofunctionalized titanium surface is positively charged, attracting more negatively charged proteins and osteoblasts to the titanium surface [12,15]. Photofunctionalization can also significantly reduce the carbon content of aged titanium surfaces [49,50]. The reduction in hydrocarbons aids cell adhesion and promotes cell proliferation, thereby accelerating and enhancing bone formation [11,50].

However, interestingly, contradictory results from single studies were included in this systematic review. Heo et al. found that there were no significant differences in the ISQ between photoactivated and non-photoactivated implants [32]. No significant differences in the improvement of photofunctionalized implant stability were observed, with the exception of implants placed in poor-quality bone, by Choi et al. [29]. These results can be related to limitations in methodological issues and the inadequate sample size.

In order to compare the rate of developing implant stability, Funato et al. calculated the ISQ increase per month of photofunctionalized implants, which ranged from 2.0 to 8.7 depending on the ISQ at placement, and compared the data with those of untreated implants reported in the previous articles [21]. It was found that the ISQ increase per month of photofunctionalized implants was considerably higher than that of untreated implants reported in the literature. Suzuki et al. defined the OSI as the ISQ increase per month, that is, [(ISQ at loading) − (ISQ at implant placement)]/(healing time in months), and suggested that photofunctionalized implants showed a higher OSI than the non-photofunctionalized implants reported in the literature [23]. The OSI represents the rate of developing implant stability over a period of healing time, allowing researchers to draw a comparison among implants with different healing time. The result of this meta-analysis showed that photofunctionalization significantly improved the OSI, which can be explained by the finding in the animal study that the photofunctionalized implants accomplished bone–titanium integration four times faster [15].

In terms of MBL assessment, the results showed that the difference in MBL between implants with and without photoactivated surface treatment was not statistically significant. However, only two studies were amenable to the meta-analysis [28,31]. According to Choi et al. and Sandhu et al., lower MBL of photofunctionalized implants was observed in both studies [29,30]. Unfortunately, due to the fact that MBL was measured at different points or the data of MBL and standard deviation values were not reported, their studies were not selected for the meta-analysis. Hence, there are limitations to the interpretation of this result, and further studies are needed to confirm the effect of photofunctionalization on MBL.

Existing clinical data fully demonstrate that photofunctionalization improves the rate of establishing implant stability and appears to improve implant stability and reduce MBL. Our explanation for the limited clinical role of photofunctionalization in implants stability and MBL is based on the assumptions below.

First of all, the limited number of pooled studies and methodological flaws are factors.

Secondly, the experimental design was unsatisfactory, and the available data published were incomplete. Some studies were limited to specific bone conditions and jaw regions. Three studies did not report the ISQ and its standard deviation values [27,29,30], and one study did not report MBL and its standard deviation values [29]. We have contacted the corresponding authors but haven’t got additional data.

In addition, several factors contributed to the high heterogeneity and may have had a specific impact on the study results: (1) Bone quality [5]. There were large differences in baseline bone among the included studies. (2) Implant characteristics, including implant type, diameter, length, and surface treatment [5,6]. (3) The parameters of photofunctionalization. Differences in contact angles of water droplets on the titanium surfaces were observed for the different types of UV light used. Moreover, the titanium-mediated enhancement of osteoconductivity was substantially improved by UVC treatment but not UVA treatment [51]. The wavelength of UV radiation in the included studies varied from 250 nm to 382 nm [22,27,28,29,31], and the processing time of UV irradiation varied from 10 min to 20 min in six studies [24,27,28,29,30,31].

In view of the above limitations, there are some suggestions for future research. Firstly, more RCTs with large-scale and long-term follow-up are needed to further verify the long-term clinical effects of photofunctionalization through strict experimental design, excluding the impacts of bone quality, implant characteristics, and other hybrid factors on implant prognosis. Secondly, it is necessary to conduct RCTs to compare the effects of UV treatments with different wavelengths and times on bone integration, implant stability, and MBL, so as to find the parameters of photofunctionalization with the best clinical effects.

## 5. Conclusions

Based on the positive effect of photofunctionalization on the rate of establishing implant stability, photofunctionalization may provide an effective and practical strategy to achieve faster osseointegration and reduce the overall healing time. Photofunctionalization appears to improve the implant stability, particularly in poor-quality bone or in complex cases requiring staged or simultaneous site-development surgery. However, the clinical effect of photofunctionalization on MBL remains unclear due to the shortage of available studies. Further high-quality trials are needed to supplement reliable evidence for the clinical effects of photofunctionalization on implants.

## Figures and Tables

**Figure 1 jcm-11-07042-f001:**
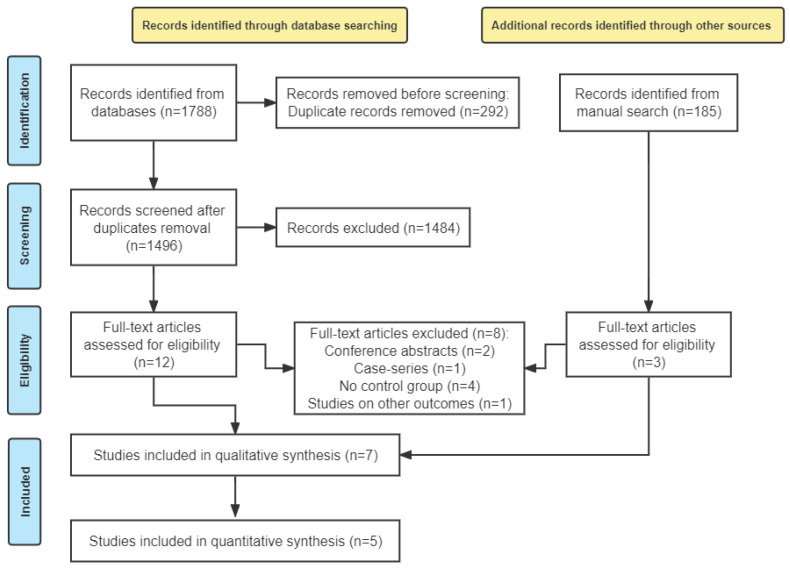
PRISMA flow chart of study selection for the systematic review and meta-analysis.

**Figure 2 jcm-11-07042-f002:**
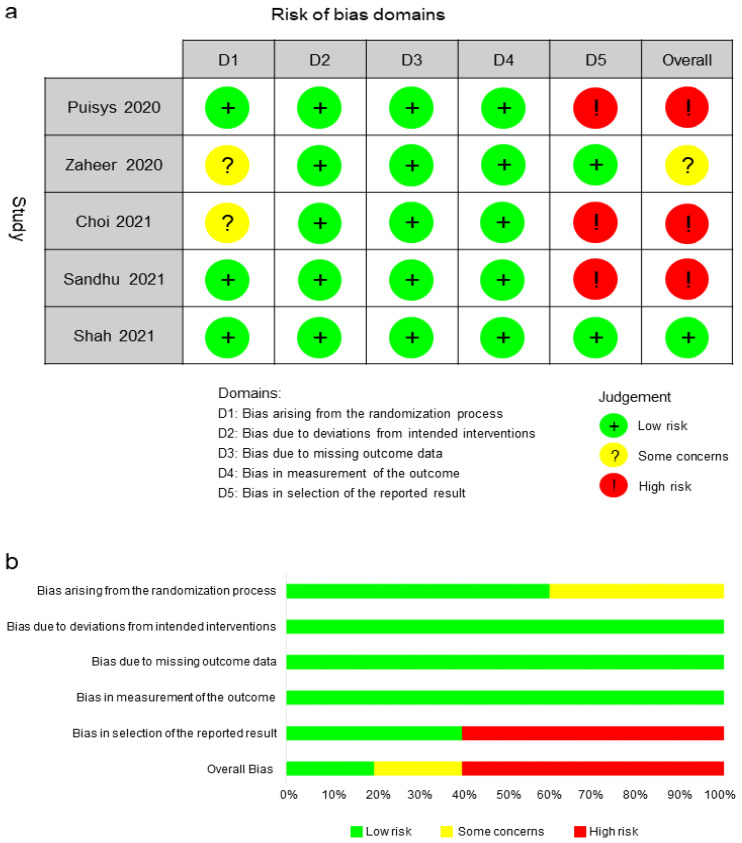
Risk-of-bias assessment using ROB 2.0: (**a**) risk-of-bias summary and (**b**) risk-of-bias graph [27,28,29,30,31].

**Figure 3 jcm-11-07042-f003:**
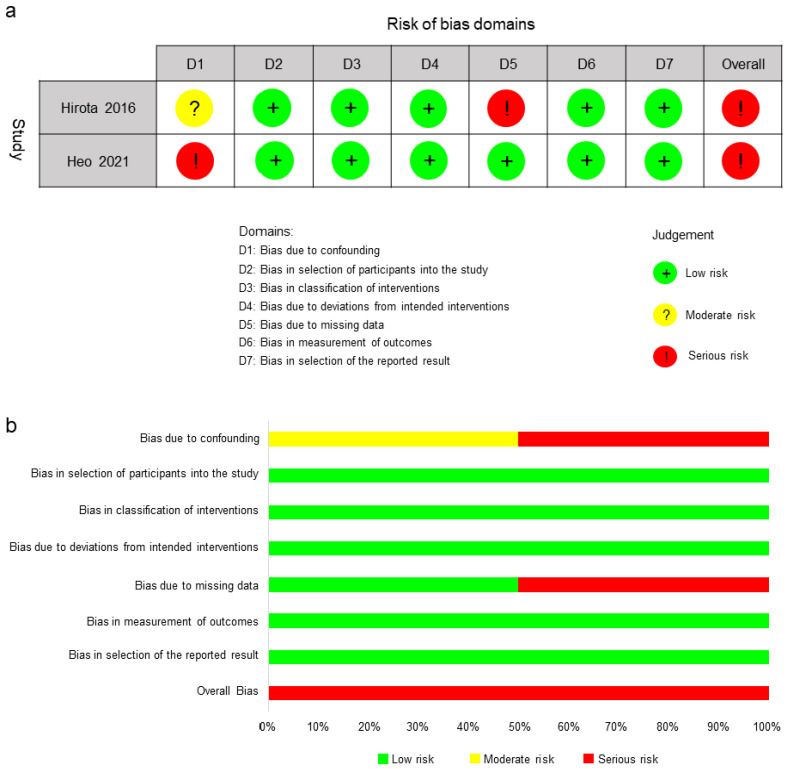
Risk-of-bias assessment using ROBINS-I tool: (**a**) risk-of-bias summary and (**b**) risk-of-bias graph [24,32].

**Figure 4 jcm-11-07042-f004:**
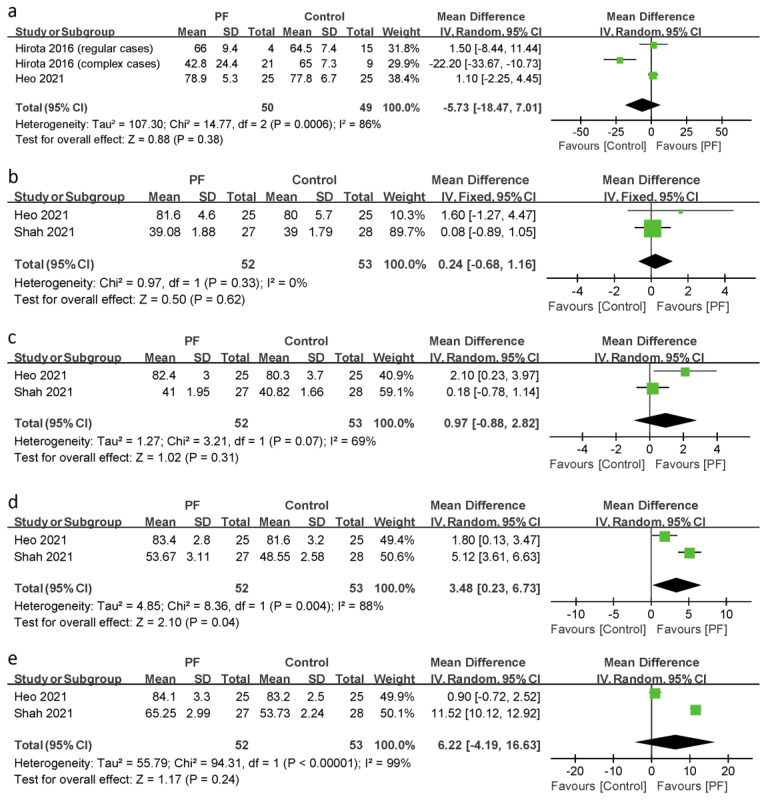
Forest plot of implant stability: (**a**) immediately, and (**b**) 2 weeks, (**c**) 4 weeks, (**d**) 2 months, and (**e**) 4 months after the placement of the implants. The rhombuses represent pooled results. PF = photofunctionalization; CI = confidence interval; SD = standard deviation [24,31,32].

**Figure 5 jcm-11-07042-f005:**
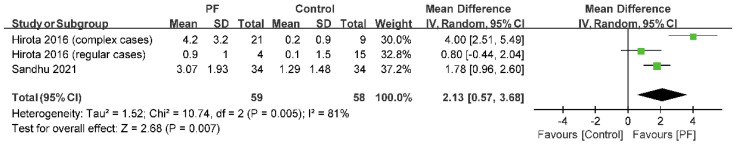
Forest plot of OSI. The rhombus represents pooled results. PF = photofunctionalization; Legend: CI = confidence interval; SD = standard deviation [24,30].

**Figure 6 jcm-11-07042-f006:**
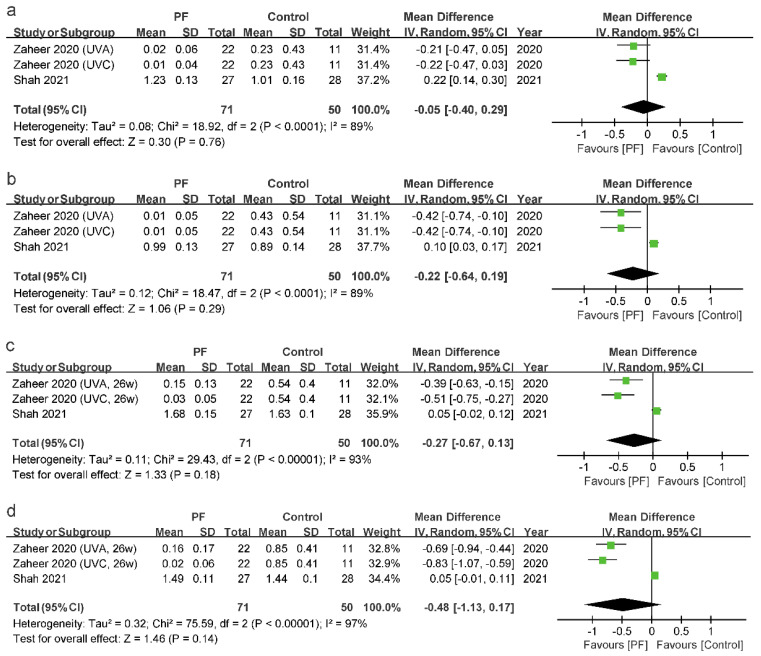
Forest plot of MBL: (**a**) on the mesial side of the implants at 2 months, (**b**) on the distal side of the implants at 2 months, (**c**) on the mesial side of the implants at 6 months, and (**d**) on the distal side of the implants 6 months after the placement of the implants. The rhombuses represent pooled results. PF = photofunctionalization; Legend: CI = confidence interval; SD = standard deviation [28,31].

**Table 1 jcm-11-07042-t001:** General characteristics of the included studies.

Author (Year)	Country	Study Design	Patients	Mean Age (Range)	Gender (M/F)	Implants	Arch	Immediate Implant	Time Period of Placement—Final Reconstruction
Hirota (2016) [24]	Japan	Case-control study	7	47–63	3/4	49	Max/Mnd	Yes/No	3–8 months
Puisys (2020) [27]	Lithuania	RCT ^S^	180	50.65 (24–78)	69/111	360	Max/Mnd	No	NR
Zaheer (2020) [28]	Pakistan	RCT	66	40.6 ± 12.1	24/42	66	Max/Mnd	No	NR
Choi (2021) [29]	Korea	RCT	34	66.13 (32–88)	15/19	57	Max	No	4 months
Sandhu (2021) [30]	India	RCT ^S^	34	46.94 ± 12.03	20/14	68	Max/Mnd	No	3 months
Shah (2021) [31]	India	RCT	84	50.7 ± 7.1	47/37	84	Max	Yes	6 months
Heo (2021) [32]	Korea	CCT	25	63.9	12/13	50	Max/Mnd	No	NR

CCT, clinical controlled trial; F, female; M, male; Max, maxilla; Mnd, mandible; NR, not reported; RCT, randomized controlled trial; ^S^ split-mouth design.

**Table 2 jcm-11-07042-t002:** Characteristics of the included studies pertaining to photofunctionalization.

Author (Year)	Photofunctionalization Device	Wavelength (nm)	Photofunctionalization Time (min)
Hirota (2016) [24]	TheraBeam Affiny (Ushio)	A mixture of spectra ^TA^	15
Puisys (2020) [27]	TheraBeam^®^ SuperOsseo Device (Ushio Inc., Sazuchi Bessho-cho, Himejij, Hyogo, Japan)	A mixture of spectra ^TS^	12
Zaheer (2020) [28]	UV ACUBE 100 (Honle, Grafelfing, Germany)	UVA, 382	10
		UVC, 260	10
Choi (2021) [29]	TheraBeam Affiny (Ushio Inc., Tokyo, Japan)	A mixture of spectra ^TA^	15
Sandhu (2021) [30]	Lelesil Innovative Systems (Thane, India)	NR	15
Shah (2021) [31]	Ultraviolet rays chamber (SK Dent)	253.7	20
Heo (2021) [32]	NR	NR	NR

NR, not reported; UV, ultraviolet; UVA, ultraviolet A; UVC, ultraviolet C; ^TA^ λ = 360 nm and λ = 250 nm; ^TS^ intensity = 0.05 mW/cm^2^ (λ = 360 nm) and intensity = 2 mW/cm^2^ (λ = 250 nm).

**Table 3 jcm-11-07042-t003:** Outcomes of the included studies.

Author (Year)	Main Findings	Conclusions
Hirota (2016) [24]	The average OSI and the OSI in complex cases were considerably greater for photofunctionalized implants than for as-received implants. Photofunctionalized implants showed significantly higher ISQ2 values than as-received implants.	Photofunctionalization accelerated the rate and enhanced the final level of implant stability development compared with as-received implants, particularly for implants placed into poor-quality bone and other complex cases.
Puisys (2020) [27]	At 2, 3, 4, and 8 weeks, the RT values were higher in photoactivated implants than those in control implants, being statistically significant.	The photoactivation of the surface of titanium implants improved healing and implant stability, especially in the early healing phase.
Zaheer (2020) [28]	Both UVA- and UVC-treated groups showed minimal MBL compared with control group, with no significant differences between the two experimental groups.	Photofunctionalized SLA-coated titanium dental implants showed positive biological response after the healing phase in contrast to the non-UV-treated group.
Choi (2021) [29]	In bone quality group III (grayscale value between 300 and 500), significant differences were observed in terms of the differences in the resonance frequency analysis values 4 weeks and 4 months postoperatively. In bone quality group II (grayscale value above 500), the UV-treated group showed significantly lesser bone loss 4 weeks postoperatively.	UV surface treatment on implants may increase the initial stability in the region of the maxilla with poor bone quality.
Sandhu (2021) [30]	The PF group showed a statistically significantly higher OSI than the control group. Statistically significantly higher crestal bone loss was observed in the control group as compared with the PF group.	Implants with photofunctionalized surfaces reduced overall healing time and crestal bone loss. Photofunctionalization was an effective aid for achieving faster osseointegration with good crestal bone stability.
Shah (2021) [31]	Mean MBL was not significantly different between the PF group and the control group. The PF group showed significantly greater implant stability than the control group.	Pretreatment of commercial dental implants with PF exhibited a statistically significant difference in implant stability but not in other outcomes.
Heo (2021) [32]	The photoactivated implants showed higher ISQ values than those without photoactivated surface treatment. However, there were no significant differences between the two implant groups.	The photoactivated implant surface appeared to have higher implant stability than that without photoactivation by increasing the hydrophilic surface.

ISQ, implant stability quotient; ISQ2, ISQ values measured at stage-two surgery; MBL, marginal bone loss; OSI, osseointegration speed index; PF, photofunctionalization; RT, removal torque; UVA, ultraviolet A; UVC, ultraviolet C.

## Data Availability

Please see the original included articles.

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
