# Peer review of "Clinical Effects of Photofunctionalization on Implant Stability and Marginal Bone Loss: Systematic Review and Meta-Analysis"

_jcm, 2022, doi:10.3390/jcm11237042_

Round 1

Reviewer 1 Report

I enjoy reading this systematic review. It answers a question of recent interest. The following parameters are set by manufacturer (Therabeam® SuperOsseo, Ushio, Tokyo, Japan) for optimum titanium UV exposure for 12 min; intensity was 0.05 mW/cm2 (λ = 360 nm) and 2 mW/cm2 (λ = 250 nm). Perhaps if the authors can check if the authors follow the setting set by the manufacturers? If this is confirmed, this information can be presented  as a subscript to the table

Author Response

Response to Reviewer 1 Comments

We sincerely thank you for your carefulness and conscientiousness, and we sincerely appreciate your constructive and helpful comments. According to your suggestions, we have finished a revision marked up using the “Track Changes” function and resubmitted. The main corrections in the paper and the responds to your comments are as follows:

Point: The following parameters are set by manufacturer (Therabeam® SuperOsseo, Ushio, Tokyo, Japan) for optimum titanium UV exposure for 12 min; intensity was 0.05 mW/cm2 (λ = 360 nm) and 2 mW/cm2 (λ = 250 nm). Perhaps if the authors can check if the authors follow the setting set by the manufacturers? If this is confirmed, this information can be presented as a subscript to the table.

Response: Thanks for your constructive suggestion. Except that two RCTs specifically demonstrated that the wavelength of UV radiation was 382 nm (UVA), 260 nm (UVC) and 257.3 nm, other studies included in this systematic review didn’t report the specifical wavelength. The implants in two studies were treated with UV irradiation using the photo device TheraBeam Affinity Device (Ushio Inc., Japan), and the implants in another RCT were subjected to UV irradiation using the photofuntionalization device, TheraBeam® SuperOsseo Device (Ushio Inc., Hyogo, Japan) as per the manufacturer’s recommendation. We have checked the parameters set by the manufacturers. TheraBeam Affinity Device delivered UV light as a mixture of spectra via single source of UV lamp at λ = 360 nm and λ = 250 nm [1], and the UV light of TheraBeam® SuperOsseo Device was generated as a mixture of spectra; intensity was about 0.05 mW/cm2 (λ = 360 nm) and 2 mW/cm2 (λ = 250 nm) [2]. We have modified Table 2 and added the above information to Table 2.

Table 2. Characteristics of the included studies pertaining to photofuntionalization

Author (year)

Photofunctionalization device

Wavelength

(nm)

Photofunctionalization time (minutes)

Hirota 2016 [22]

TheraBeam Affiny, Ushio

a mixture of spectraTA

15

Puisys 2020 [29]

TheraBeam® SuperOsseo Device (Ushio Inc., Sazuchi Bessho-cho, Himejij, Hyogo, Japan)

a mixture of spectraTS

12

Zaheer 2020 [30]

UV ACUBE 100 (Honle, Germany)

UVA,382

10

UVC,260

10

Choi 2021 [31]

TheraBeam Affiny; Ushio Inc., Tokyo, Japan

a mixture of spectraTA

15

Sandhu 2021 [23]

Lelesil Innovative Systems, Thane, India

NR

15

Shah 2021 [24]

ultraviolet rays chamber (SK Dent)

253.7

20

Heo 2021 [32]

NR

NR

NR

NR, not reported; UV, Ultraviolet; UVA, Ultraviolet A; UVC, Ultraviolet C; TA λ = 360 nm and λ = 250 nm; TS intensity = 0.05 mW/cm2 (λ = 360 nm) and intensity = 2 mW/cm2 (λ = 250 nm)

References

[1] Tuna, T.; Wein, M.; Swain, M.; Fischer, J.; Att, W. Influence of Ultraviolet Photofunctionalization on the Surface Characteristics of Zirconia-Based Dental Implant Materials. Dent. Mater. 2015, 31, e14–e24, doi:10.1016/j.dental.2014.10.008.

[2]   Henningsen, A.; Smeets, R.; Hartjen, P.; Heinrich, O.; Heuberger, R.; Heiland, M.; Precht, C.; Cacaci, C. Photofunctionalization and Non-Thermal Plasma Activation of Titanium Surfaces. Clin. Oral Investig. 2018, 22, 1045–1054, doi:10.1007/s00784-017-2186-z.

Reviewer 2 Report

Dear authors,

Firstly, congratulations for a great manuscript and extraordinary work. The manuscript titled: „The clinical effects of photofunctionalization on implant stability and marginal bone loss: a systematic review and meta-analysis“ is very well written and is backed up with appropriate number of references referring to previous studies.

Abstract and is well concised and focused on the background.

I suggest to extend a little bit an introduction part and add the aim in the end of this section.

Materials and methods are explained in detail and discussion is well presented.

Regarding conclusion part I suggest to rewrite this because it is too general.

Author Response

Response to Reviewer 2 Comments

We sincerely thank you for reviewing our manuscript and your positive comments on our manuscript, and we sincerely appreciate your valuable and constructive comments. In accordance with your suggestions, we have finished a revision marked up using the “Track Changes” function and resubmitted. The main corrections in the paper and the responds to your comments are as follows:

Point 1: I suggest to extend a little bit an introduction part and add the aim in the end of this section.

Response 1: Thanks for your constructive suggestion. We have added some content in the introduction section. Some recent studies on photofunctionalization have been further introduced. In addition, we have added the aim of this systematic review in the end of this section.

Point 2: Regarding conclusion part I suggest to rewrite this because it is too general.

Response 2: Thank you for your constructive comments. Based on your comments, in our resubmitted manuscript, the conclusion has been revised into - Based on the positive effect of photofunctionalization on the rate of establishing implant stability, photofunctionalization may provide an effective and practical strategy to achieve faster osseointegration and reduce the overall healing time. Photofunctionalization appears to improve the implant stability, especially in poor-quality bone or in complex cases requiring staged or simultaneous site-development surgery. However, the clinical effect of photofunctionalization on MBL remains unclear due to the shortage of available studies. Further high-quality trials are needed to supplement reliable evidence for the clinical effects of photofunctionalization on implants.

Reviewer 3 Report

I appreciate the authors' idea, the study is interesting and up-to-date. The study meets the scientific requirements and is well documented. I think it can be a starting point for other research in this direction.

I have some questions/recommendations :

1.     In chapter 2.2.PICOS criteria- There is no restriction on gender- do you think that age cannot be an independent variable that can influence the study?

2.     In chapter 3.2.- didn't you find studies from 2016 to 2020? Maybe I didn't understand exactly what you were referring to.

3.     Your conclusion is - Photofunctionalization increases the rate of establishing implant stability, aids to achieve faster osseointegration and appears to improve the implant stability- I think you shouldn't be so categorical, taking into account the contradictory studies mentioned in the discussions. I think it should be appreciated as positive of  photofunctionalization but the limits or independent variables shoul be also mentioned.

4.     The references chapter does not fully respect the recommended style

5.     Are you thinking of continuing your research and if so, in what way?

Author Response

Response to Reviewer 3 Comments

We feel great thanks for your professional review work and your positive comments on our manuscript, and we sincerely appreciate your constructive comments and suggestions. According to your comments, we have finished a revision marked up using the “Track Changes” function and resubmitted. The main corrections in the paper and the responds to your comments are as follows:

Point 1: In chapter 2.2. PICOS criteria- There is no restriction on gender- do you think that age cannot be an independent variable that can influence the study?

Response 1: Thanks for your comment. We perceive that age can be a variable that can influence the outcomes of the dental implants. Firstly, we made the PICOS criteria- Patients (P): aged at least 18 years. The studies won’t meet the PICOS criteria of this systematic review and won’t be included if there are patients younger than 18 years. Secondly, we consider age as a baseline characteristic. In the process of methodological quality assessment, we checked whether there is a substantial excess in statistically significant differences in baseline characteristics, including age between intervention group and control group. If the trial has large baseline imbalances, it will be likely to be judged in moderate or serious risk using the Cochrane risk-of-bias tool 2 (RoB 2.0) or the ROBINS-I tool. We found that there is no significant difference in the age of patients between intervention group and control group in five RCTs. The case-control study and the CCT included in our study did not report the age of intervention group and control group, respectively. Thus, the age can be considered as a baseline confounding factor in the case-control study and the CCT, which were both judged in serious risk.

Point 2: In chapter 3.2.- didn't you find studies from 2016 to 2020? Maybe I didn't understand exactly what you were referring to.

Response 2: Thank you very much for your comment. We conducted electronic searches in the databases without publication date restriction on 4 September 2022. According to the eligibility criteria, seven studies were included in this systematic review. One out of the seven included studies was published in 2016, two studies were published in 2020, and four studies were published in 2021.

There may be some ambiguity in this sentence and thank you for pointing this out. This sentence has been revised into - Among the seven included studies, one study was published in 2016, two were published in 2020 and the other four were published in 2021.

Point 3: Your conclusion is - Photofunctionalization increases the rate of establishing implant stability, aids to achieve faster osseointegration and appears to improve the implant stability- I think you shouldn't be so categorical, taking into account the contradictory studies mentioned in the discussions. I think it should be appreciated as positive of  photofunctionalization but the limits or independent variables shoul be also mentioned.

Response 3: Thank you for your constructive comments. Although there is a significant increase in stability measured at 2 months (P=0.04; MD=3.48; 95% CI=-0.23 to 6.73) and OSI (I2=81%; P=0.007; MD=2.13; 95% CI=0.57 to 3.68) of photofunctionalized implants, contradictory results from single studies were also included in this systematic review. Based on your comments, in our resubmitted manuscript, the conclusion has been revised into - Based on the positive effect of photofunctionalization on the rate of establishing implant stability, photofunctionalization may provide an effective and practical strategy to achieve faster osseointegration and reduce the overall healing time. Photofunctionalization appears to improve the implant stability, especially in poor-quality bone or in complex cases requiring staged or simultaneous site-development surgery. However, the clinical effect of photofunctionalization on MBL remains unclear due to the shortage of available studies. Further high-quality trials are needed to supplement reliable evidence for the clinical effects of photofunctionalization on implants.

Point 4: The references chapter does not fully respect the recommended style.

Response 4: We sincerely thank you for careful reading and we are sorry for our carelessness. In our resubmitted manuscript, the references chapter has been revised according to MDPI Citations Style Guide. Thanks for your correction.

Point 5: Are you thinking of continuing your research and if so, in what way?

Response 5: Thanks for your comment. Photofunctionalization is one of the recent techniques of implant surface modification. It has been proved that photofunctionalization can improve the efficiency and capacity of protein adsorption, decrease the surface hydrocarbon content and increase hydrophilicity [1,2,3,4].

Based on the qualitative and quantitative assessment of clinical effects of photofunctionalization in this systematic review, it can be concluded that photofunctionalization may provide an effective and practical strategy to achieve faster osseointegration. Photofunctionalization appears to improve the implant stability. However, the clinical effect of photofunctionalization on MBL remains unclear due to the shortage of available studies. Considering some limitations of the existing clinical trials, we plan to implement a RCT to eliminate the interference of hybrid factors as far as possible through strict experimental design. In particular, we would like to conduct a RCT for patients with poor-quality bone or in complex cases requiring staged or simultaneous site-development surgery to determine whether photofunctionalization is a reliable method to improve the osseointegration process and establish long-term success in complex cases.

In addition, we think it will also be a research direction to study whether the parameters of UV radiation will affect the clinical effects of photofunctionalization. In vitro experiments, differences in contact angles of water droplets on the titanium surfaces were observed for the different types of UV light used. Moreover, the titanium-mediated enhancement of osteoconductivity was substantially improved by UVC treatment but not UVA treatment [5]. However, to the best of our knowledge, only one clinical trial has studied whether UVA and UVC have different effects on MBL of dental implants. Whereas considering the limitations of the devices, it may be not easy to carry out an clinical research on the influence of different wavelengths of UV on dental implants.

References:

[1] Hori, N.; Ueno, T.; Suzuki, T.; Yamada, M.; Att, W.; Okada, S.; Ohno, A.; Aita, H.; Kimoto, K.; Ogawa, T. Ultraviolet Light Treatment for the Restoration of Age-Related Degradation of Titanium Bioactivity. Int. J. Oral Maxillofac. Implants 2010, 25, 49–62.

[2] Hori, N.; Ueno, T.; Minamikawa, H.; Iwasa, F.; Yoshino, F.; Kimoto, K.; Lee, M.C.-I.; Ogawa, T. Electrostatic Control of Protein Adsorption on UV-Photofunctionalized Titanium. Acta Biomater. 2010, 6, 4175–4180, doi:10.1016/j.actbio.2010.05.006.

[3] Att, W.; Ogawa, T. Biological Aging of Implant Surfaces and Their Restoration with Ultraviolet Light Treatment: A Novel Understanding of Osseointegration. Int. J. Oral Maxillofac. Implants 2012, 27, 753–761.

[4] Gajiwala, M.; Paliwal, J.; Husain, S.Y.; Dadarwal, A.; Kalla, R.; Sharma, V.; Sharma, M. Influence of Surface Modification of Titanium Implants on Improving Osseointegration: An in Vitro Study. J. Prosthet. Dent. 2021, 126, 405.e1-405.e7, doi:10.1016/j.prosdent.2021.06.034.

[5] Gao, Y.; Liu, Y.; Zhou, L.; Guo, Z.; Rong, M.; Liu, X.; Lai, C.; Ding, X. The Effects of Different Wavelength UV Photofunctionalization on Micro-Arc Oxidized Titanium. PLoS ONE 2013, 8, e68086, doi:10.1371/journal.pone.0068086.
